# Dietary Crude Fiber Levels for Optimal Productivity of Male Ross 308 Broiler and Venda Chickens Aged 1 to 42 Days

**DOI:** 10.3390/ani12101333

**Published:** 2022-05-23

**Authors:** Muzi Ginindza, Khanyisile R. Mbatha, Jones Ng’ambi

**Affiliations:** 1School of Interdisciplinary Research and Graduate Studies, College of Graduate Studies, University of South Africa, P.O. Box 392, Pretoria 0003, South Africa; mbathkr@unisa.ac.za; 2Department of Agricultural Economics and Animal Production, University of Limpopo, Private Bag X1106, Sovenga 0727, South Africa; jones.ngambi@ul.ac.za

**Keywords:** indigenous, growth rate, neutral detergent fiber digestibility

## Abstract

**Simple Summary:**

Crude fiber levels that are not controlled in chicken diets could have adverse effects on productivity. The use of alternative protein resources in commercial broiler and indigenous chicken production often has limitations in terms of dietary crude fiber. Evidence suggests that there might be differences in the way chicken breeds cope with dietary crude fiber, which ultimately affects the production potentials. The study determined the levels of dietary crude fiber that enhanced the productivity in Ross 308 broiler and indigenous Venda chickens. Results showed that both chicken breeds aged 1 to 42 days required dietary crude fiber for enhanced productivity. However, indigenous Venda chickens required higher dietary CF levels in order to improve the production performance than the Ross 308 broiler chickens. Thus, it was concluded that the indigenous chickens were able to cope when challenged with higher dietary crude fiber levels.

**Abstract:**

The study determined the effects of dietary crude fiber (CF) levels on the production performance of male Ross 308 broiler and indigenous Venda chickens. There were 360 male Ross 308 broiler and male Venda chickens aged 1–21 for Phase 1 and 196 chickens aged 22–42 days for Phase 2. Chickens were allocated four diets with different levels of CF (3, 4, 5, and 7%) in a completely randomized design. Quadratic regression analyses were used to determine the dietary CF levels for the optimal production responses. In Phase 1, the feed intake, growth, live weight, nitrogen retention, and metabilizable energy (ME) intake of the Ross 308 broiler chickens were improved at dietary CF levels of 3.9, 4.5, 4.5, 3.2, and 3.7%, respectively. In the Venda chickens, feed intake, growth, live weight, nitrogen retention, and ME intake were optimized at 4.4, 4.8, 4.7, 4.1, and 3.3% CF, respectively. In Phase 2, the feed intake, nitrogen retention, and neutral detergent fiber digestibility (NDFD) were enhanced at CF levels of 6.4, 4.4, and 3.7% in the Ross 308 broiler chickens, respectively. Dietary CF levels of 4.5, 5.8, 5.7, 5.1, 3.9, and 4.4% optimized the feed intake, growth rate, live weight, nitrogen retention, NDF, and ADFD, respectively, in Venda chickens. It was concluded that the indigenous Venda chickens coped better with higher dietary CF than the Ross 308 broiler chickens.

## 1. Introduction

Feed costs in poultry account for a high proportion of the cost of production. The high feed cost is associated with the used protein sources [1]. Usually, feed producers tend to use fishmeal and or soybean in southern Africa as the main source of protein in poultry diets. Alternative sources of protein are therefore needed to promote the sustainability and productivity of enterprises [2]. Other protein sources such as leaf meals and agricultural by-products may be used, however, their utilization is limited because of the anti-nutritional factors including more crude fiber (CF) content.

The presence of high dietary CF limits the gut performance and thus modulates digestion [3]. While it is not practical to produce diets that have no CF, diets for commercial broiler chickens are often formulated to contain minimal dietary CF levels [4]. This minimal CF in diets is controlled because chickens do not have digestive enzymes to degrade fiber to extract nutrients such as ruminants. However, the practice of minimally including dietary CF has been criticized by other authors [5,6] because its functions are not clearly understood. This increases the bulk of the gut digesta to give it form for enhanced motility [7]. Optimal dietary fiber may improve nutrient digestibility [8,9] and the level depends on whether an animal is monogastric or ruminant.

Dietary CF provides substrates for the gut microbiota to maintain gastrointestinal tract function and health [10]. However, poultry diets are composed of a wide range of plant-based products and grains (cereal and pulses), implying significant levels of the cell wall or cellulose content. Thus, chickens need to be able to contend with diets with varied levels of CF. The challenge is that information on the appropriate dietary CF levels for optimum performance is not conclusive. 

The indigenous Venda chickens have been reported to have a higher tolerance to dietary CF [11] compared to commercial chickens. Indigenous chickens are hardy and adaptive [12], but the extent of CF digestibility is inconclusive. Production of these chickens is usually under subsistent systems [13,14]. Their diet may be characterized by seasonal supplementation with grains, but are usually are left to scavenge for food on their own. Other reports [15,16] have investigated the chickens’ crop contents and observed that the dietary CF level ranged between 4 and 12%. The information on the optimal tolerance to dietary CF levels between commercial broiler and indigenous chickens is limited. More information in this regard may help the poultry producers of these chickens formulate diets that are effective in achieving higher productivity. 

The objective of this study was to determine the effects of dietary crude fiber levels on the production performance of male Ross 308 broiler and indigenous Venda chickens aged 1 to 42 days.

## 2. Materials and Methods

### 2.1. Study Site

The study was conducted at the University of Limpopo Livestock Unit (latitude: 23°53′9.60″ S, longitude: 29°44′16.80″ E). The study procedures were reviewed and approved by the University of Limpopo Animal Research Ethics Committee (AREC/04/2018: PG). The experiments were conducted in two phases: Phases 1 and 2 when the chickens were 1 to 21 days and 22 to 42 days of age, respectively. The poultry house, by design, was open on one side with curtains.

#### 2.1.1. Phase 1

A total of 320 (160 male Ross 308 broiler and 160 male Venda chickens) day old chicks were offered experimental diets containing similar levels of crude protein (230 g/kg DM) and energy (12 MJ metabolizable energy/kg DM), which met the minimum nutrient requirements for growth productivity [17,18]. The experimental diets had four levels of dietary CF 3, 4, 5, or 7% and similar experimental diets were fed to both the 160 male Ross 308 broiler chickens and 160 male Venda chickens (Table 1). Thus, we employed a 2 (breeds) × 4 (experimental diets) factorial arrangement in a completely randomized design. Each of the four treatments had four groups and each group had 10 chickens that were put in pens measuring 1.5 m^2^. The floor pens had wood shavings as litter. All of the chickens were vaccinated against Newcastle, infectious bronchitis, and Gumboro disease. Feed (in the form of a mash) and water were made available *ad libitum* while lighting was provided throughout the phase. Curtains were closed while controlling internal temperature by heating using infra-red heat lamps.

#### 2.1.2. Phase 2

This phase commenced with 192 male chickens (96 male Ross 308 broiler and 96 male Venda chickens) aged 22 days, different to those used in Phase 1. These chickens were raised between 1 to 21 days on a starter diet containing 233 g CP/kg DM, 14 MJ GE and a CF level of 4%. The experimental diets fed from age 22–42 days were formulated to contain varying levels of CF (3, 4, 5, or 7%) while being isonitrogenic and isocaloric (220 g CP/kg DM and 12.4 MJ ME/kg DM, respectively) (Table 1). These diets were formulated to meet the minimum nutrient requirements of these chickens [17,18]. A 2 (breeds) × 4 (dietary CF levels) factorial arrangement in a completely randomized design was used. Each of the treatments had three groups and each group had eight chickens confined in a pen measuring 1.5 m^2^. The floor pens had wood shavings as litter. Feed in mash form and water were made available ad libitum. Curtains were opened and closed in the mornings and evenings, respectively, during this phase for ventilation and temperature control.

### 2.2. Data Collection

#### 2.2.1. Phase 1

Feed intake was determined by subtracting the weight of feed refusals from the total feed given daily for each pen. All of the chickens (Ross 308 broiler and Venda chickens) were weighed at the start of the experimental phase using an electronic sensitive weighing balance (Radwag, PS 4500/C/2); thereafter, weekly measurements were taken until day 21. The average body weight for each bird per pen was calculated by dividing the total weight of the birds by the number of birds in each pen. The growth rate per bird per day was calculated by subtracting the initial average body weight from the final average body weight, which was divided by the number of days per week. 

The health status and mortality of chickens in all of the pens were checked daily. The mortality rate was calculated by dividing the sum of deaths by the number of live chickens in each pen and multiplied by 100. 

#### 2.2.2. Phase 2

Feed intake was determined as described in the data collection of Phase 1. The average weight for each bird per pen was calculated by dividing the sum of the weights of individual birds in each pen divided of the number of birds per pen. The growth rate and per bird per day was calculated by subtracting the initial average body weight from the final average body weight, which was divided by the number of days per week

### 2.3. Digestibility Trial

The digestibility trial was conducted when the chickens were aged 14 to 21 and 35 to 42 days old in Phases 1 and 2, respectively. One chicken per replicate was transferred into a metabolic cage equipped with feeder and water troughs. A four-day acclimatization period was allowed before a daily fecal collection period of three days commenced. Feces voided by each bird were collected at 09.00 and weighed daily. Care was taken to avoid contamination from feathers, feed, scales, and other debris. Both samples of feed and feces were analyzed for nutrient composition for the determination of nutrient digestibility. The following equation was used [19]:(1)$$\mathrm{Nutrient}\;\mathrm{digestibility}\;(\%)=\frac{\left(\mathrm{Nutrient}\;\mathrm{consumed}\;-\;\mathrm{Nutrient}\;\mathrm{voided}\right)}{\mathrm{Nutrient}\;\mathrm{consumed}}\;\times \;100$$

Nitrogen retention was determined using the following equation [20]:(2)$$\mathrm{Nitrogen}\;\mathrm{retention}\;(\%)=\frac{\mathrm{Nitrogen}\;\mathrm{intake}\;-\mathrm{Nitrogen}\;\mathrm{excretion}}{\mathrm{Nitrogen}\;\mathrm{intake}}\;\times \;100$$
where:(3)Nitrogen intake=intake (g/d) × diet nitrogen (% DM)
(4)Nitrogen excretion=Excretion (g/d) × fecal nitrogen (% DM)

Metabolizable energy intake (MJ) was calculated by multiplying the metabolizable energy (MJ/kg) of the feed by intake for the experimental period (kg) [21]. The metabolizable energy was determined as follows:(5)$$\mathrm{Metabolizable}\;\mathrm{energy}=\frac{\mathrm{Feed}\;\mathrm{gross}\;\mathrm{energy}\;-\;\mathrm{Excretion}\;\mathrm{gross}\;\mathrm{energy}}{\mathrm{Feed}\;\mathrm{intake}}$$

### 2.4. Chemical Analysis

The dry matter, NDF, ADF, ash, and CP contents of the diets, refusals, and feces samples were determined as described by AOAC [22]. The gross energy of the diets and excreta samples was determined using an oxygen bomb calorimeter at the University of Limpopo, South Africa.

### 2.5. Statistical Analyses

The normality of the distribution of the data was checked using Shapiro–Wilk test. All data from both phases of the experiment on the productivity and nutrient digestibility of chickens were analyzed using the general linear model procedure of the SAS 9.3 package [21]. The following model was used for both trials:Y = µ + Brd + CF + (Brd × CF) + e(6)
where Y = observation; µ = overall mean; Brd = breed effect; CF = dietary crude fiber level; Brd× dietary CF = interaction effect; e = random error.

Fisher’s least significant difference (LSD) test was applied for the mean separation where there were significant differences (*p* < 0.05). The responses in the production performance, nutrient digestibility, nitrogen retention, and ME intake to the dietary CF level were modeled using the following quadratic equation:(7)$${Y=a+b}_{1}x+\;b_{2}x^{2}+e$$
where Y = production performance, nutrient digestibility, nitrogen retention, and ME intake; a = intercept; b_1_ and b_2_ = coefficients of the quadratic equation; x = dietary crude fiber level; –b_1_/2b_2_ = x value for optimal response. The quadratic model was fitted to the experimental data by means of the NLIN procedure of SAS [23].

The linear relationships between the dietary CF level and responses in the production performance and digestibility were modeled using the following linear equation:Y = a + bx + e(8)
where Y = production performance and digestibility; a = intercept; b = coefficient of the linear equation; x = dietary CF level.

## 3. Results

### 3.1. Phase 1

The nutrient composition of diets offered to the male Ross 308 broiler and Venda chickens aged 1 to 21 days had increased (*p* < 0.05) values of NDF and ADF (Table 2).

Dietary CF levels had effects (*p* < 0.05) on the performance in terms of the measured traits of both the Ross 308 broiler and Venda chickens (Table 3). The male indigenous Venda chickens tolerated higher dietary CF levels of 4.4, 4.8, and 4.7% for the enhanced feed intake, growth rate, and live weight, respectively, than in the Ross 308 broiler chickens (3.9, 4.5, and 4.5%, respectively) (Table 4). A 5% dietary CF level resulted in an improved feed conversion ratio (FCR) in the Venda chickens while no effect was observed in the male Ross 308 broiler chickens. However, the metabolizable energy intake was optimized at a higher dietary CF level of 3.7% in the Ross 308 broiler chickens than the 3.3% dietary CF of Venda chickens. 

Nitrogen retention was enhanced at different dietary CF levels of 3.2 and 4.1% in the Ross 308 broiler and indigenous Venda chickens, respectively (Table 4). The dietary CF level and NDFD as well as the ADFD had linear and negative correlations in both chicken breeds aged 1 to 21 days (Table 5). There were, however, breed differences wherein the Venda chickens had a higher (*p* < 0.05) NDFD and ADFD while the feed intake, growth rate, and live weight of the Ross 308 broiler chickens were higher (*p* < 0.05) than the Venda chickens. Mortality was not affected (*p* > 0.05) by the dietary CF level in the Ross 308 broiler and Venda chickens.

### 3.2. Phase 2

The feed intake, growth rate, nitrogen retention, NDFD, and live body weight were affected (*p* < 0.05) by the dietary CF level in the male Ross 308 broiler chickens aged 22 to 42-days old. However, these production variables and the ADFD of the male Venda chickens were affected (*p* < 0.05) by the dietary CF level (Table 6). The feed intake of the male Ross 308 broiler chickens was the only variable optimized at a higher dietary CF level (6.4%) than in the male Venda chickens (4.5% dietary CF) (Table 7). However, nitrogen retention and NDFD were enhanced at a higher dietary CF level (5.1 and 3.9%, respectively) in the male Venda chickens than in the male Ross 308 broiler chickens with 4.4 and 3.7% for the nitrogen retention and NDFD, respectively.

There were negative and strong correlations between the dietary CF level and growth rate, FCR, and live body weight of the male Ross 308 broiler chickens (Table 8). However, the dietary CF levels of 5.8, 6.4, and 5.7% improved the growth rate, FCR, and live body weight, respectively, in the male Venda chickens.

## 4. Discussion

The experimental diets had CF fiber levels ranging from 3 to 7%. There was no control diet with 0% CF because varied levels of crude fiber are inadvertently incorporated into the diet when mostly plant-based ingredients are utilized [24].

### 4.1. Phase 1 

The dietary CF level had effects on the production variables (feed intake, growth rate, nitrogen retention, metabolizable energy intake, NDFD, ADFD, and live body weight) of both the male Ross 308 broiler and Venda chickens aged 1 to 21 days. Hetland et al. [3], Choct et al. [6], and Salami and Odunsi [25] reported consistent findings with the current study, where varying the dietary CF affected the productivity of young chickens regardless of the breed. This is essentially due to limitations in dealing with the fiber fraction, which differs not only within animal species but also within breeds. In the current study, the Ross 308 broiler chickens had a lower dietary CF level for the optimal feed intake compared to that of the Venda chickens. Gonzalez-Alvarado et al. [26] reported that increased feed particle size (due to dietary CF) may result in ingesta remaining in the proventriculus and gizzard for a long time. This slows the passage rate of the ingested food, resulting in a lowered feed intake [27].

The growth rate and live body weight of the male Venda chickens were improved at a higher dietary CF level than the male Ross 308 broiler chickens. Jimenez-Moreno et al. [8] reported a lower dietary CF level for the optimal growth rate and live weight of 3.4% in broiler chickens aged 1 to 21 days, while Alabi et al. [17] reported a higher dietary CF level of 5.5% for the enhanced growth rate and live weight for the Venda chickens raised in closed confinement. Broiler chickens are selectively bred for high growth performance, thus feed that is nutrient dense facilitates better performance [28]. Dietary CF levels used in the study did not affect the FCR of the male Ross 308 broiler chickens aged 1 to 21 days. This was perhaps due to the proportionate changes in the feed intake and growth rate parameters. Sarikhan et al. [4] and Adibmoradi et al. [29] reported contrary findings to the current study and reported that dietary CF levels of 3 to 3.9% optimized the FCR in young broiler chickens.

A higher dietary CF level of 3.7% improved the ME intake in the male Ross 308 broiler chickens compared to 3.3% in the male Venda chickens. Adibmoradi et al. [29] found no effect of the inclusion level of insoluble fiber on the ME intake of broiler chicks. However, chickens can control their feed intake depending on the limiting nutrients of a particular diet [30,31]. Furthermore, Mbajiorgu et al. [32] reported that broiler chickens had a better ability to control their voluntary feed intake than indigenous chickens. This may have been the case in the current study where the increased dietary CF level tended to dilute the nutrient concentration. Hence, a higher dietary CF level optimized the ME intake in the broiler chickens than in the Venda chickens. Nitrogen retention was affected by the dietary CF level in both chicken breeds. The Ross 308 broiler chickens were optimized at a lower dietary CF level (3.2%) than the Venda chickens (4.1%). Therefore, the Ross 308 broiler chickens were more sensitive to the level of fiber, which tended to limit the nitrogen utilization. Similar findings were reported by Jorgensen et al. [33], who also showed that an increased dietary CF level resulted in decreased nitrogen retention in the Ross 308 broiler chickens. These authors suggested that the reduction in nitrogen retention resulted from the chickens feeding less, and higher endogenous nitrogen losses incurred when offered diets high in CF. There have been limited studies that have investigated the nitrogen retention in indigenous Venda chickens.

Both chicken breeds had linear and negative relationships with the ADFD and NDFD from 1 to 21 days of age. Thus, increases in the dietary CF level resulted in the decreased NDFD and ADFD of the male Ross 308 broiler and Venda chickens. Results of the present study, however, showed that the Venda chickens had a higher NDFD and ADFD compared to the Ross 308 broiler chickens. Chickens do not produce enzymes that are essential for the digestion of celluloses and hemicelluloses found in their diet, thus they depend on the limited microbial fermentation in the lower gut [6,34,35]. It is possible that there might be different microbial compositions in the indigenous Venda chickens that are capable of fiber digestion, resulting in a higher NDFD and ADFD than the Ross 308 broiler chickens. This, however, needs to be validated.

### 4.2. Phase 2

Varying dietary CF levels had effects on the performance traits of both the male Ross 308 broiler and Venda chickens aged 22 to 42 days. A dietary CF level of 6.4% for the optimal feed intake in male Ross 308 broiler chicken was higher than the 4.5% dietary CF observed in the male Venda chickens. Since broiler chickens can adjust their voluntary feed intake, more of the fibrous diet may have been consumed in order to attempt to tolerate the limiting dietary protein and energy. Thus, the feed intake was optimized at a higher dietary CF level in the Ross 308 broiler chickens than in the Venda chickens. 

The growth rate, FCR, and live weight of the Ross 308 broiler chickens were adversely affected by the increased dietary CF levels. Adibmoradi et al. [29], Gonzalez-Alvarado et al. [26], and Amerah et al. [36] reported contrary findings in broiler chickens where dietary CF levels of 4.5% optimized the growth rate and live weight of the broiler chickens aged 22 to 42 days. However, the dietary CF levels of 5.8 and 5.7% enhanced the growth rate and live weight, respectively, in the male Venda chickens. These values were higher than those reported to have an optimized growth performance in broiler chickens by other authors [26,29]. Indigenous chickens might have capabilities enabling them to thrive and utilize fibrous diets. These mechanisms merit further investigations.

Nitrogen retention was optimized at a higher dietary CF level (5.1%) in the Venda chickens than in Ross 308 broiler chickens (4.4%). Jorgensen et al. [33] observed that increased dietary CF level negatively affected the nitrogen retention in broiler chickens. These authors suggested that when broiler chickens are faced with higher dietary CF levels, they tend to eat less and suffer higher endogenous nitrogen losses. These factors are critical in influencing nitrogen retention in broiler chickens. 

Indigenous Venda chickens had higher NDFD than the Ross 308 broiler chickens. This may have been possible because of the presence of specific cellulose degrading micro-organisms in their gut [37]. Therefore, this gives an advantage to indigenous Venda chickens when challenged with diets high in CF. Schokker et al. [38] and Kers et al. [39] reported that the genetic makeup of chickens is recognized as an important factor that determines the intestinal microbial composition. Further studies investigating the enzyme activities and microbial composition in indigenous Venda chickens are needed to elucidate their fiber digesting capabilities.

## 5. Conclusions

In Phases 1 and 2, the dietary CF level influenced the productivity of both the male 308 broiler and male Venda chickens. The feed intake, growth, and live weight of the male Ross 308 broiler chickens aged 1 to 21 days were optimized at lower dietary CF levels compared with the male Venda chickens while the NDFD and ADFD were adversely affected by increases in the dietary CF level in both chicken breeds. 

In Phase 2 (22–42 days), the feed intake of the male Ross 308 broiler chickens was improved at a higher dietary CF level (6.4%) than the 4.1% in the male Venda chickens while the growth rate and live weight of the male Ross 308 broiler chickens decreased with higher dietary CF levels. However, in the male Venda chickens, the growth rate and live weight were enhanced at dietary CF levels of 5.8 and 5.7%, respectively. Furthermore, the NDFD was optimized at a higher dietary CF level in the male Venda chickens than in the male Ross 308 broiler chickens. Therefore, this indicates that the male Ross 308 broiler chickens were more affected by the dietary CF level while the indigenous Venda chickens had an ability to cope with high fiber diets. Further studies are necessary to investigate the coping mechanisms of indigenous chickens when fed high CF diets.

## Figures and Tables

**Table 1 animals-12-01333-t001:** Ingredients (%) of the experimental diets used in Phases 1 and 2.

Ingredients	Phase 1				Phase 2			
	% CF				% CF			
	3	4	5	7	3	4	5	7
Maize	49.4	49.2	50.2	50	54.4	51.9	50.1	50
Maize gluten	7.9	4.7	4.7	13.6	3.9	4.7	6.7	13.5
Soyabean meal	28.0	29.6	27.3	12.3	28.0	27.6	24.3	12.3
Wheat bran	0.0	2.0	2.0	2.0	0.0	2.3	2.1	2.4
Maize bran	3.0	1.7	7.1	11.4	0.0	1.9	5.1	11.4
Potato protein	2.0	5	1.1	2.0	2.0	2.0	2.0	2.0
Disodium phosphate	0.2	0.0	0.0	0.0	0.2	0.0	0.0	0.0
Calcium carbonate	3.8	0.0	0.0	0.0	3.8	0.0	0.0	0.0
Salt	0.2	0.3	0.3	0.3	0.2	0.3	0.3	0.3
Dicalcium phosphate	1.8	4	4.1	4.2	1.8	4.0	4.1	4.2
Sodium bicarbonate	0.2	0	0	0.1	0.2	0	0	0.1
DL methionine	0.2	0.1	0.1	0.1	0.2	0.1	0.1	0.1
L-lysine HCl	0.2	0.1	0.1	0.1	0.2	0.1	0.1	0.1
L-threonine	0.1	0.1	0.1	0.1	0.1	0.1	0.1	0.1
Vit-min premix ^1^	1	1	1	1	1	1	1	1
Sunflower oil	2	2.2	1.9	2.8	4	4	4	2.8

^1^ Supplied per kilogram diet: iron (ferrous sulfate), 60 mg; manganese (manganese sulfate and manganese oxide), 120 mg; zinc (zinc oxide), 100 mg; iodine (calcium iodate), 1 mg; copper (copper sulfate), 8 mg; selenium (sodium selenite), 0.3 mg, vitamin A, 9600 IU; vitamin D3 3600 IU; vitamin E, 18 mg; vitamin B12, 15 μg; riboflavin, 10 mg; niacin, 48 mg; D-pantothenic acid, 18 mg; vitamin K, 2 mg; folic acid, 1.2 mg; vitamin B6, 4 mg; thiamine, 3 mg; D-biotin, 72 μg.

**Table 2 animals-12-01333-t002:** Nutrient composition (g/kg) (mean ± standard error, on DM basis) of the experimental diets fed to the male Ross 308 broiler and Venda chickens during Phases 1 and 2.

Nutrient	Phase 1				Phase 2			
	Diet CF Level				Diet CF Level			
	3	4	5	7	3	4	5	7
DM	892 ^a^ ± 1.8	882 ^a^ ± 1.2	905 ^a^ ± 4.2	900 ^a^ ± 1.0	912 ^a^ ± 10.5	922 ^a^ ± 10.7	915 ^a^ ± 6.8	900 ^a^ ± 12.1
CP	230 ^a^ ± 0.5	230 ^a^ ± 0.4	230 ^a^ ± 0.4	230 ^a^ ± 0.6	219 ^a^ ± 2.1	221 ^a^ ± 4.2	218 ^a^ ± 3.1	218 ^a^ ± 3.4
ME (MJ/kg)	12.0 ^a^ ± 0.4	12.0 ^a^ ± 0.8	12.0 ^a^ ± 0.5	12.0 ^a^ ± 0.0	12.4 ^a^ ± 0.43	12.4 ^a^ ± 0.25	12.4 ^a^ ± 0.10	12.4 ^a^ ± 0.05
NDF	88.4 ^d^ ± 2.6	91.0 ^c^ ± 8.7	116.7 ^b^ ± 3.6	133.1 ^b^ ± 9.1	86.4 ^d^ ± 5.31	97.3 ^c^ ± 3.41	126.8 ^b^ ± 6.75	147.3 ^a^ ± 10.32
ADF	22.1 ^d^ ± 2.1	28.2 ^c^ ± 1.0	34.0 ^b^ ± 2.4	42.8 ^b^ ± 1.2	24.3 ^d^ ± 3.13	31.2 ^c^ ± 2.91	42.8 ^b^ ± 4.77	53.9 ^a^ ± 5.13
Ash	77 ^a^ ± 0.8	77 ^a^ ± 0.3	76 ^a^ ± 0.8	79 ^a^ ± 0.9	82 ^a^ ± 2.1	78 ^a^ ± 3.7	84 ^a^ ± 4.4	72 ^a^ ± 7.3

^a, b, c, d:^ Means in a column having different superscripts were significantly different (*p* < 0.05); CF: Crude fiber; CP: Crude protein; GE: Gross energy; NDF: Neutral detergent fiber; ADF: Acid detergent fiber.

**Table 3 animals-12-01333-t003:** The effect of dietary crude fiber level on the production performance of the male Ross 308 broiler and Venda chickens aged 1 to 21 days.

Treatment			Variable *						
		Feed Intake (g/d)	Growth Rate (g/d)	Nitrogen Retention (%)	FCR	ME Intake (MJ)	NDFD (%)	ADFD (%)	Live Weight (g)
CF (%)									
Ross 308	3	46 ^a^ ± 0.41	29.0 ^a^ ± 0.51	90.0 ^a^ ± 2.52	1.6 ^c^ ± 0.11	12.1 ^a^ ± 0.65	32.9 ^c^ ± 4.12	26.1 ^a^ ± 2.13	650 ^a^ ± 24.0
	4	46 ^a^ ± 0.52	30.2 ^a^ ± 0.41	80.0 ^a^ ± 1.19	1.5 ^c^ ± 0.10	12.2 ^a^ ± 0.85	33.2 ^c^ ±4.51	26.0 ^a^ ± 1.31	675 ^a^ ± 25.1
	5	46 ^a^ ± 0.61	30.3 ^a^ ± 0.32	89.4 ^a^ ± 1.45	1.5 ^c^ ± 0.13	11.9 ^a^ ± 0.50	27.7 ^c^ ± 1.61	20.0 ^b c^ ± 2.32	678 ^a^ ± 23.0
	7	39 ^b^ ± 0.82	26.7 ^b^ ± 0.52	72.2 ^b^ ± 3.13	1.5 ^c^ ± 0.12	10.5 ^b^ ± 0.45	25.1 ^d^ ± 0.32	18.0 ^b c^ ± 2.41	600 ^b^ ± 17.2
Venda	3	26 ^d^ ± 0.41	6.9 ^d^ ± 0.34	61.4 ^c^ ± 3.41	3.8 ^a^ ± 0.33	9.4 ^c^ ± 0.17	36.3 ^b^ ± 1.02	27.4 ^a^ ± 0.61	375 ^c^ ± 20.4
	4	29 ^c^ ± 0.51	9.8 ^c^ ± 0.31	72.9 ^c^ ± 5.55	3.0 ^b^ ± 0.20	9.7 ^c^ ± 0.30	38.7 ^a^ ± 0.84	26.5 ^a^ ± 0.41	517 ^a^ ± 10.3
	5	26 ^d^ ± 0.32	7.5 ^d^ ± 0.42	60.0 ^c^ ± 6.52	3.6 ^a^ ± 0.30	8.8 ^c d^ ± 0.70	34.7 ^b c^ ± 0.6	24.2 ^b^ ± 0.67	395 ^c^ ± 18.5
	7	25 ^d^ ± 0.91	6.9 ^d^ ± 0.33	45.7 ^d^ ± 4.94	3.6 ^a^ ± 0.29	7.2 ^d^ ± 0.95	31.6 ^c^ ± 2.70	19.2 ^c^ ± 0.97	374 ^c^ ± 26.1
Breed									
Ross 308		44.3 ^a^ ± 0.6	29.1 ^a^ ± 0.41	82.9 ^a^ ± 3.73	1.5 ^b^ ± 0.12	11.7 ^a^ ± 0.40	29.7 ^b^ ± 3.21	22.5 ^b^ ± 1.98	651 ^a^ ± 24.3
Venda		26.5 ^b^ ± 0.5	7.8 ^b^ ± 0.52	60.0 ^b^ ± 3.36	3.5 ^a^ ± 0.31	8.8 ^b^ ± 0.64	35.0 ^a^ ±1.17	24.3 ^a^ ± 0.84	415 ^b^ ± 50.1
*Probabilities*									
CF level		0.0350	0.0419	0.0413	0.0419	0.0501	0.0407	0.0149	0.0312
Breed		0.0010	0.0017	0.0482	0.0017	0.0243	0.0041	0.0415	0.0017
CF * breed interactions		0.0783	0.1326	0.6671	0.1326	0.1741	0.0561	0.0628	0.0722

^a, b, c, d^: Means in a column having different superscripts were significantly different (*p* < 0.05); *: Mean ± standard error; CF: Crude fiber; ME: Metabolizable energy; NDFD: Neutral detergent fiber digestibility; ADFD: Acid detergent fiber digestibility.

**Table 4 animals-12-01333-t004:** The dietary crude fiber levels for the optimal production performance of the male Ross 308 broiler and Venda chickens aged 1 to 21 days.

Variable	Formula	r^2^	CF level (%)	Optimal Y-Level	Probability
Male Ross 308 Broiler chickens					
Feed intake	Y = 34.164 + 6.268X − 0.795X^2^	0.988	3.9	46.5	0.001
Growth rate	Y = 17.955 + 5.498X − 0.607X^2^	0.991	4.5	30.4	0.001
N-retention	Y = 77.17 + 6.45X − 1.01X^2^	0.791	3.2	87.5	0.047
ME intake	Y = 10.076 + 1.142X − 0.155X^2^	0.999	3.7	12.2	0.00
Live weight	Y = 412.564 + 118.118X − 13.045X^2^	0.991	4.5	679.9	0.002
Male Venda chickens					
Feed intake	Y = 19.891 + 3.405X − 0.386X^2^	0.466	4.4	34.8	0.067
Growth rate	Y = 0.142 + 3.549X − 0.373X^2^	0.388	4.8	8.6	0.164
FCR	Y = 5.469 − 0.865X + 0.086X^2^	0.264	5.0	3.3	0.378
N retention	Y = 23.72 + 21.06X − 2.58X^2^	0.826	4.1	66.7	0.042
ME intake	Y = 7.725 + 1.097X − 0.168X^2^	0.965	3.3	11.1	0.067
Live weight	Y = 81.273 + 157.114X − 16.659X^2^	0.335	4.7	908.6	0.074

r^2^: Coefficient of determination; CF: Crude fiber.

**Table 5 animals-12-01333-t005:** The relationships between the dietary crude fiber level (%) and nutrient digestibility (%) of the male Ross 308 broiler and Venda chickens aged 14–21 days.

Variable	Formula	r	Probability
Male Ross 308 broiler chickens			
NDFD	Y = 40.080 − 2.180X	−0.934	0.045
ADFD	Y = 33.206 − 2.249X	−0.925	0.052
Male Venda chickens			
NDFD	Y = 42.260 − 1.460X	−0.837	0.016
ADFD	Y = 34.409 − 2.123X	−0.987	0.013

r: Correlation coefficient; NDFD: Neutral detergent fiber digestibility; ADFD: Acid detergent fiber digestibility.

**Table 6 animals-12-01333-t006:** The effect of the dietary crude fiber level on the production performance of the male Ross 308 broiler and Venda chickens aged 22 to 42 days.

Treatment			Variable *						
		Feed Intake (g/d)	Growth (g/d)	FCR	Nitrogen Retention (%)	ME Intake (MJ)	NDFD (%)	ADFD (%)	Live Weight (g)
CF (%)									
Ross 308	3	149.5 ^b^ ± 0.30	77.8 ^b^ ± 0.27	1.91 ^d^ ± 0.142	76.5 ^c^ ± 2.16	12.0 ^a^ ± 0.31	44 ^c d^ ± 0.2	32 ^b^ ± 1.2	2095 ^a^ ± 12.1
	4	148.9 ^b^ ± 6.12	88.7 ^a^ ± 4.23	1.69 ^d^ ± 0.166	88.2 ^a^ ± 2.12	11.9 ^a^ ± 0.06	46 ^c^ ± 0.1	34 ^b^ ± 0.6	2217 ^a^ ± 90.5
	5	163.0 ^a^ ± 1.44	68.8 ^c^ ± 0.13	2.37 ^b^ ± 0.269	79.4 ^b^ ± 1.32	11.7 ^a^ ± 0.03	43 ^c d^ ± 1.9	32 ^b^ ± 1.7	1899 ^b^ ± 45.7
	7	160.5 ^a b^ ± 4.64	68.3 ^c^ ± 1.35	2.35 ^b^ ± 0.441	64.7 ^d^ ± 3.08	11.8 ^a^ ± 0.10	41 ^d^ ± 1.6	30 ^b^ ± 0.6	1895 ^b^ ± 21.1
Venda	3	82.1 ^d^ ± 0.08	22.8 ^e^ ± 2.04	3.66 ^a^ ± 0.355	63.3 ^e^ ± 0.15	11.9 ^a^ ± 0.1	41 ^d^ ± 0.3	32 ^b^ ± 0.9	666 ^d^ ± 28.8
	4	84.9 ^c^ ± 0.38	28.0 ^e^ ± 2.39	3.03 ^a^ ± 0.262	66.7 ^d^ ± 1.09	11.5 ^a^ ± 0.1	48 ^b^ ± 0.9	39 ^a^ ± 0.1	794 ^c d^ ± 29.6
	5	85.1 ^c^ ± 0.09	37.1 ^d^ ± 2.71	2.2 ^c^ ± 0.46	80.0 ^b^ ± 2.08	12.0 ^a^ ± 0.0	52 ^a^ ± 1.4	32 ^b^ ± 0.5	904 ^c^ ± 73.2
	7	76.0 ^e^ ± 0.18	34.8 ^d^ ± 2.31	2.18 ^c^ ± 0.17	63.1 ^e^ ± 2.21	11.2 ^a^ ± 0.1	39 ^d^ ± 1.7	37 ^a b^ ± 0.3	849 ^c^ ± 73.6
Breed									
Ross 308		155.5 ^a^ ± 3.66	75.9 ^a^ ± 4.80	2.15 ^b^ ± 0.132	77.2 ^a^ ± 4.1	2.6 ^a^ ± 0.2	44 ^a^ ± 1.0	32 ^a^ ± 0.8	2026 ^a^ ± 79
Venda		82.0 ^b^ ± 2.13	30.7 ^b^ ± 3.16	2.82 ^a^ ± 0.339	68.3 ^b^ ± 2.3	2.0 ^b^ ± 0.1	45 ^a^ ± 3.0	35 ^a^ ± 1.8	803 ^b^ ± 51.0
*Probabilities*									
CF level		0.0496	0.0006	0.0484	0.0328	0.0622	0.0012	0.0006	0.0059
Breed		0.0001	0.0015	0.0277	0.0452	0.0508	0.2757	0.8212	0.0001
CF level X breed		0.0776	0.0647	0.0743	0.5214	0.3214	0.1745	0.0174	0.2374

^a, b, c, d, e^: Means in a column with different superscripts were significantly different (*p* < 0.05); *: Mean ± standard error; FCR: Feed conversion ratio; CF: Crude fiber; NDFD: Neutral detergent fiber digestibility; ADFD: Acid detergent fiber digestibility.

**Table 7 animals-12-01333-t007:** The dietary crude fiber levels for the optimal production performance of the male Ross 308 broiler and Venda chickens aged 22 to 42 days.

Variable	Formula	r^2^	CF Level (%)	Optimal Y-Level	Probability
Ross 308 Broiler chickens					
Feed intake	Y = 112.99 + 14.96X − 1.16X^2^	0.680	6.4	161.2	0.055
N-retention	Y = 27.295 + 25.78X − 2.93X^2^	0.867	4.4	84.0	0.037
NDFD	Y = 40.127 + 2.486X − 0.341X^2^	0.748	3.7	44.7	0.032
Venda chickens					
Feed intake	Y = 55.04 + 13.51X − 1.503X^2^	0.980	4.5	85.4	0.005
Growth rate	Y = −25.22 + 21.13X - 1.807X^2^	0.957	5.8	36.6	0.003
N-retention	Y = −10.94 + 34.12X − 3.345X^2^	0.744	5.1	76.1	0.051
NDFD	Y = −18.291 + 28.245X − 2.864X^2^	0.938	4.9	51.3	0.004
ADFD	Y = 30.145 + 1.377X − 0.068X^2^	0.110	4.4	37.1	0.157
Live weight	Y = −199.25 + 388.50X − 34.06X^2^	0.993	5.7	908.6	0.008

r^2^: Coefficient of determination; CF: Crude fiber.

**Table 8 animals-12-01333-t008:** The relationships between the dietary crude fiber level (%) and production performance of the male Ross 308 broiler chickens aged 22 to 42 days.

Variable	Formula	r	Probability
Growth rate	Y = 93.181 − 3.638X	−0.647	0.0353
Feed conversion ratio	Y = 1.471 + 0.147X	0.922	0.017
Live weight	Y = 2346 − 67.4X	−0.897	0.0506

r: Correlation coefficient.

## Data Availability

Not applicable.

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
