# Peer review of "Dietary Crude Fiber Levels for Optimal Productivity of Male Ross 308 Broiler and Venda Chickens Aged 1 to 42 Days"

_animals, 2022, doi:10.3390/ani12101333_

Round 1

Reviewer 1 Report

The aim of this study was to determine the effect of dietary crude levels on the production performance of male Ross 308 broiler and indigenous Venda chickens in rearing period. The results obtained are important for producers of chicken. The Introduction and Discussion chapters are well written, others need to be supplemented or changed. The list of proposed changes is presented below:

General comments:

Please prepare a research paper according to the instructions for authors

E-mail addresses; The initials of the first and last names (the same as in the Author Contributtions chapter) are to be completed

Table 1-6 headers must be in bold

Notations "p" for significance with lowercase letter in italic, space before and after the "<"

In Reference chapter, volume number must be in italic, page range long dash from the insert function

Detailed comments:

L24 Conect sentences with the next paragraph

L74 22 to 42 days or 1 to 42 days?

Phase 1 add information about the type of building (closed, with windows?), Type of bedding, microclimate (temperature, humidity, lighting program). Add the same information for Phase 2

L89 4 subgroup, 10 male chickens

L90 Gumboro disease; 1.5 m2, space after the number

L96 233 g, 14.0 MJ, space after the number

L101 three subgroup, each subgroup

The sum of the ingradients for the individual diets must be 100%. Currently it is 100.3%; 99.9%, 100.1%, 100.1% for Phase 1

Provide information about the form of the diet (fine) and the frequency of its administration,

L172 + What statistical characteristics are provided, what test was used to verify the normal distribution, and what test was used to verify the significance of differences

In table 2 the „data” in one line

L192 add explanations for ME, NDFD, ADFD

In table 2 line under Diet CF level, Diet CF level instead of% CF

In table 3 and 4 Item * instead of Variable, remove "*" in the title

L225 add an explanation for the NDFD and ADFD

Reviewer 2 Report

Due to the limiting availability of cereals and protein sources and the increasing feed prices, in the future more locally available feedstuffs, industrial by-products will be used in animal nutrition. Since most of the alternative feedstuffs contain higher fibre, more knowledge needed on the fibre requirement and tolerance of the different farm animal species. For poultry no real requirements and tolerance levels have been published. So, the topic of the paper is novel. However, in the past decades it has also been proved, that crude fibre and neutral detergent fibre recover only a variable part of the fibre fraction of diets and therefore unfit to evaluate the fibre fractions in feedstuffs and poultry diets. Beside structural fibre the soluble fractions play also crucial role in nutrient digestibility, gut viscosity and gut health. So, using only CF levels cannot explain the responses in the measured parameters.

In the materials and methods part, it should be explained what nutrient requirements have been used for the diets and the predicted ME contents should be given instead of gross energy.

Regarding the diet compositions, the main changes in the diets were the decreasing soybean meal and increasing maize gluten and maize bran with the increase of the CF content. It means not only the CF but also the amino acid compositions of the diets have changed. The diets with higher CF contained for example contained less lysine and more leucine.  In spite this, the crystalline amino acid supplementation was almost the same in all diets. What could be the reason for this? Beside the composition the measured nutrient contents of the diets should also be given. Otherwise, it is difficult to decide that the results are related only to the genotypes and CF contents.

In the digestibility trial it is strange that the fibre fractions digestibilities have been measured only. The fibre digestion capacity of the chicken is limited and the variation between individual birds is high. So, the question is why not fat and starch digestibility was measured beside or instead of NDF and ADF? In the digestibility trial one chicken per replicate was used. It means 4 replicates, which is not really enough according to the experiences of the reviewer.

In the chemical analysis part, more detailed description of the analytical methods and equipments should be given. It is not clear what the averages and SD values in Table 2 mean. This table contains the measured nutrient contents. What was the replicate in this analysis comparison? As it has been mentioned before more measured nutrient contents needed (crude fat, starch, sugar or N-free extract, amino acids) and the calculated ME value form the crude nutrients.

Among production traits FCR is missing. It can be found in the footer of table 6, but the results are missing.

The 21-day old live weight of the Venda birds at the 4% CF treatment is very high, 517 g, compared with the other groups. If Venda birds consumed less feed and retained significantly less N, how could be the live weight of this group close to the Ross chickens. If the daily growth rate of Venda birds was around 7-10 g/day, the body weight at day 21 should be around 7 x 21 = 150 g. What is the reason of these discrepancies?

Since the N-retention of animals was determined with the balance method, the retention should be given also in %.  

In table 4 and 5 the regressions are only quadratic in the case of production traits, N retention and ME, and only linear in the case of NDF and ADF digestibility. What was the reason for that? Was in the first group only quadratic responses significant and only linear in the second? Please tell also the significance of the regression equations beside the r2 values.

In table 6, the final body weight of the birds is too low. Male Ross chickens can reach around 3 kg live weight at day 42. What could be the reason for this poor growth rate even at the control diet. Comparing with literature data the ADF and NDF digestibility values are too high. These fibre fractions contain only insoluble components, mainly cellulose, hemicellulose and lignin and due to their low solubility, their absorption and bacterial fermentation is low. What could be the reason, that the cellulose and lignin content of the diets were absorbed at more than 30%?

Round 2

Author Response

Comments from Reviewer 1

Response

I agree that CF and the van Soest methods are official methods for fibre analysis, however if you want to determine the optimal fibre content of the diets the soluble fraction can not be ignored. Mostly the soluble fractions can be fermented in the caeca and can be degraded. In this trial only wheat bran contained soluble arabinoxylan, but its incorporation rate at 2% probably did not have significant impact on diet level.

Noted. Further studies in the NSPs (soluble and insoluble) would indeed give an in-depth analysis of the fibre digestion and justifications for the use exogenous enzymes in the diets of chickens. Fibre is a “heterogeneous mixture of several types of polysaccharides”.  In this study, our analyses and methods (NDF and ADF) were targeting and limited to the insoluble fractions of NSPs and compared the efficiencies between breeds.

Among production traits FCR is still missing. You can calculate is easily. So please insert these values into table 3. “Optimal productivity”, as you mention in the title of the article, cannot be evaluated without FCR.

Addressed. Now included

There is no answer in your response to this question: “The 21-day old live weight of the Venda birds at the 4% CF treatment is very high, 517 g, compared with the other groups. If Venda birds consumed less feed and retained significantly less N, how could be the live weight of this group close to the Ross chickens. If the daily growth rate of Venda birds was around 7-10 g/day, the body weight at day 21 should be around 7 x 21 = 150 g. What is the reason of these discrepancies?”

The birds of the 4% treatments were indeed heavier than the other groups of the same breed. Interestingly, whlile the results of the Growth Rate do not correspond fully with the final live weights, one must appreciate that the calculations were analysed independently, and that the calculations were determined through averages of replicates determined) weekly. Hence, they seem under-estimated.

Regarding the N retention, your explanation is not correct. The daily N retention is influenced by the feed intake. The feed intake values are given in Table 2. So, N-retention, given in % is more useful for the comparison of the CF and genotype effects.

Addressed. Values converted and analysis checked/re-ran.

Nitrogen retention in Ross chickens was more efficient than in Venda chickens. And these values tended to decrease in older birds.

In tables 4 and 5 please insert also P-values.

Addressed. Now included

In your response missing the answer to this remark: Comparing with literature data the ADF and NDF digestibility values are too high. What could be the reason, that the NDF and ADF digestion were more than 30 and 20% respectively?

The digestibility figures obtained were relative to their respective nutrient intake hence they may not compare well numerically with others in literature because the circumstances or environmental factors are different. What they do show is a comparison of fibre digestion efficiencies between broilers and indigenous chickens